# Evaluation and Projection of Surface PM_2.5_ and Its Exposure on Population in Asia Based on the CMIP6 GCMs

**DOI:** 10.3390/ijerph191912092

**Published:** 2022-09-24

**Authors:** Ying Xu, Jie Wu, Zhenyu Han

**Affiliations:** 1National Climate Center, China Meteorological Administration, Beijing 100081, China; 2Laboratory for Climate Studies, China Meteorological Administration, Beijing 100081, China; 3School of Geography and Environmental Engineering, Gannan Normal University, Ganzhou 341000, China

**Keywords:** CMIP6, PM_2.5_, projection, population exposure

## Abstract

This paper evaluates the historical simulated surface concentrations of particulate matter small than 2.5 µm in diameter (PM_2.5_) and its components (black carbon (BC), dust, SO_4_, and organic aerosol (OA)) in Asia, which come from Coupled Model Intercomparison Project Phase 6 (CMIP6). In addition, future projected changes of surface PM_2.5_ and its components, as well as their exposure to population, under the different Shared Socioeconomic Pathway (SSP) scenarios are also provided. Results show that the simulated spatial distribution of surface PM_2.5_ concentrations is consistent with the Modern-Era Retrospective Analysis for Research and Applications version 2 (MERRA-2) and Socioeconomic Data and Applications Center (SEDAC). The model spreads are small/large over the regions with low/high climatic mean surface PM_2.5_ concentrations, i.e., Northern Asia/Saudi Arabia, Iran, and Xinjiang Province of China. The multi-model ensemble of CMIP6 reproduces the main features of annual cycles and seasonal variations in Asia and its sub-regions. Under the scenarios of SSP1-2.6, SSP2-4.5, and SSP5-8.5, compared to the present-day period of 1995–2014, annual mean surface PM_2.5_ concentrations are projected to decrease in Asia, with obvious differences among the scenarios. Meanwhile, the magnitudes and timings of changes at the regional scale are quite different, with the largest decreases in South Asia (SAS). Under SSP3-7.0, the increase of surface PM_2.5_ concentrations in SAS is the largest, with the increase value of 8 μg/m^3^ in 2050; while under SSP370-lowNTCF, which assumes stronger levels of air quality control measures relative to the SSP3-7.0, the decreases of surface PM_2.5_ concentrations in SAS, East Asia (EAS) and Southeast Asia (SEAS) are the largest. The characteristics of seasonal trends are consistent with that of the annual trend. The trends in the concentrations of surface PM_2.5_ and its components are similar. The population-weighted average values of surface PM_2.5_ concentrations are projected to decrease in Central Asia (CAS), EAS, North Asia (NAS), and SEAS, and it indicates that the surface PM_2.5_ concentrations over the most populated area of Asia will decrease. In SAS, because of its large population, the impact of air pollutants on human health is still disastrous in the future. In summary, the surface PM_2.5_ concentrations over the most area of Asia will decrease, which is beneficial to air quality and human health; under SSP370-lowNTCF, the reduction of short-lived climate forcers (SLCFs) will further improve air quality.

## 1. Introduction

Air pollutants are important atmospheric constituents as they have huge impacts on human health [1]. Air quality is also a critical issue in the world at present. The impact of exposure to air pollutants on human health has been increasing over recent decades [2,3]. Among a variety of surface air pollutants, two major components are ozone (O_3_) and particulate matter less than 2.5 µm in diameter (PM_2.5_). It is estimated that exposure to these two air pollutants has caused up to 4 million premature deaths annually [4,5]. Long- and short-term exposure to ambient concentrations of surface PM_2.5_ have different degrees of correlation with some aspects of physical problem, such as mortality and cardiovascular, respiratory, metabolic, and nervous diseases [6]. In China, the total number of deaths caused by prolonged exposure to ambient concentrations of surface PM_2.5_ is about 1 to 2 million every year [7,8]. Because of the high surface PM_2.5_ concentrations and large population, China also has a high mortality burden due to short-term PM_2.5_ exposure. Li, et al. [9] reported that the premature deaths related to short-term exposure is about a seventh of that caused by long-term exposure. In terms of climate impact, O_3_/PM_2.5_ has a positive/negative radiative forcing on climate [10,11].

Previous studies have projected the surface PM_2.5_ concentrations by using global climate models. For example, Wu et al. [12] projected the surface PM_2.5_ concentrations in China will decrease in the end of 21st century, by using the Atmospheric Chemistry and Climate Model Intercomparison Project (ACCMIP) models. Wu et al. [13] showed that the surface PM_2.5_ concentrations over the Jing-Jin-Ji region will peak at a maximum in 2030–2040, using ACCMIP models. Yang et al. [14] projected the PM_2.5_ burden will decrease over most continents, especially East Asia under RCP4.5 and RCP8.5 scenarios, by using BCC_AGCM2.0_CUACE/Aero model.

The Coupled Model Intercomparison Project (CMIP) outputs a variety of climate data which are publicly available for analysis [15], thus proving the opportunity for projecting future changes in air quality. The Phase 6 of CMIP (CMIP6) includes the state-of-the-art climate global models (GCMs) [16]. The Aerosols and Chemistry Model Intercomparison Project (AerChemMIP) is endorsed by CMIP6, with the aim of documenting and understanding the changes in the chemical composition of the atmosphere. In CMIP6, a new set of emissions scenarios is used: the shared socio-economic pathways (SSPs), with the consideration to social, economic, and environmental developments [17]. Turnock et al. [18] reported that CMIP6 models underestimate the historical surface PM_2.5_ worldwide and projected the decrease of PM_2.5_ in future scenarios containing strong air quality and climate mitigation measures. Su et al. [19] evaluated CMIP6 GCMs’ skills in surface PM_2.5_ and its components over Asia in 2005–2020 and found that the spatial distributions of them can be reproduced by CMIP6 models.

Asia is a region with complex topography, complicated climate, and large population. Moreover, it ranks among the worst-polluted regions worldwide, and has the highest mortality rate caused by air pollution [4]. Therefore, it is important to understand the future change of surface PM_2.5_ and its impact on population exposure.

In this paper, we assess the performance of CMIP6 models in simulating the present surface PM_2.5_ and its components in Asia. The future changes in these air pollutants and their impact on population exposure under SSPs are also provided.

## 2. Data and Methods

### 2.1. SSP Scenarios

SSPs are updated from the Representative Concentration Pathways (RCPs) used in CMIP5. Depending on social, economic, and environmental development, the SSPs are divided into five different pathways: SSP1-sustainability, SSP2-middle of the road, SSP3-regional rivalry, SSP4-inquality, SSP5-fossil fuel development. The degree of air pollution control is assumed on top of the baseline pathway [20]. In SSP3 and SSP5, weak air pollution controls occur; in SSP2, medium controls occur, with strong controls in SSP1 and SSP5 [21]. SSP1-2.6, SSP2-4.5, and SSP5-8.5 in CMIP6 share radiative forcing levels by 2100, with RCP2.6, RCP4.5, and RCP8.5 used in CMIP5, respectively. In accordance with the Paris Agreement for keeping global temperature below 2 °C of global warming above pre-industrial levels, SSP1-1.9 is included. SSP370-lowNTCF is a pathway specific to the AerChemMIP, with the aim of studying the impact of mitigation measures, which assumes stronger levels of air quality control measures relative to the SSP3-7.0.

### 2.2. Observation Data Set

In this study, two sets of surface PM_2.5_ observations have been obtained to evaluate the CMIP6 GCMs: The Modern-Era Retrospective Analysis for Research and Applications, version 2 (MERRA-2), and the global annual PM_2.5_ grids from Socioeconomic Data and Applications Center (SEDAC).

The MERRA-2 data set assimilates observations of aerosol optical depth (AOD) from ground-based and satellite remote-sensing platforms. This observation uses the GEOS-5 atmospheric model simulations coupled to the GOCART aerosol module. In addition, separate mass mixing ratios for black carbon (BC), organic aerosol (OA), SO_4_, sea salt (SS), and dust are provided from MERRA-2. This monthly dataset spans from 1980 to 2020, with a horizontal resolution of 0.5° × 0.625° (latitude × longitude). The MERRA-2 are widely employed by many studies in assessment of aerosol simulations [18,19,22,23,24]. Previous studies reported the underestimation of PM_2.5_ over East Asia [25,26].

The SEDAC data set combines AOD obtained from multiple satellite algorithms including the NASA Moderate resolution Imaging Spectroradiometer (MODIS), Multi-angle Imaging Spectro Radiometer (MISR), MODIS Multi-Angle Implementation of Atmospheric Correction (MAIAC), and the Sea-Viewing Wide Field-of-View Sensor (SeaWiFS) [27,28]. This annual dataset covers the period from 1998 to 2019, with a horizontal resolution of 0.01° × 0.01° (latitude × longitude).

To facilitate the comparison, the observations are interpolated to a common 1° × 1° (latitude × longitude) grid, using an area-weighted interpolation algorithm [29]. Considering the temporal coverage of SEDAC, MERRA-2, and historical simulations of CMIP6, the period of 1998–2014 is selected for evaluation.

### 2.3. Surface PM_2.5_ in CMIP6

Historical monthly simulations from 12 CMIP6 GCMs (https://esgf-node.llnl.gov/search/cmip6/ (accessed on 1 March 2022)) over the period of 1850–2014 are used in this study. The monthly outputs of 8, 6, 10, 10, 8 GCMs under SSP1-2.6, SSP2-4.5, SSP3-7.0, SSP370-lowNTCF, and SSP5-8.5 scenarios are used for future projection. These simulations in SSPs were conducted over the period of 2015–2100, except for SSP370-lowNTCF, which was conducted over 2015–2055 [30]. To compare with SSP370-lowNTCF, the period of 2015–2055 in SSP3-7.0 is used in this study. Only one realization of each model is used in this study, to ensure all the GCMs are weighted equally in the multimodel statistics. Because of their different horizontal resolutions, the outputs are interpolated to a common 1° × 1° (latitude × longitude) grid, by a bilinear (for low-resolution GCMs) or an area-weighted (for high-resolution GCMs) interpolation algorithm [29].

The model information is provided in Table 1. All the models used in this study share the same anthropogenic emission inventory from the Community Emissions Data System (CEDS) [31]; but they have different schemes for simulation natural emissions aerosols, thus, with different representations of the aerosol size distribution [30].

Not all CMIP6 GCMs output surface PM_2.5_ directly, and the calculation method is not consistent among individual models due to their different treatment of aerosols. For example, the simulation of ammonium nitrate is included in only a few CMIP6 models (currently, only GISS-E2-1-G and GFDL-ESM4). Therefore, to use a consistent definition across all GCMs, we use both PM_2.5_ from the original simulations and calculated PM_2.5_ offline (the sum of BC, OA, SO_4_, SS, and dust). In this study, following Turnock et al. [18] and Su et al. [19], we assume that the sizes of all BC, OA, and SO_4_ aerosol mass is finer than 2.5 µm, with a factor of 0.25 for SS and 0.1 for dust. The formula used to estimate PM_2.5_ concentrations is expressed as
$$PM_{2.5}=BC+OA+SO_{4}+\left(0.25\times SS\right)+\left(0.1\times DUST\right)$$

The whole of Asia is the target region for analysis. To assess the PM_2.5_ changes in different regions of Asia, we divided Asia into the six sub-regions used by The Intergovernmental Panel on Climate Change (IPCC): North Asia (NAS), Central Asia (CAS), East Asia (EAS), West Asia (WAS), South Asia (SAS), and Southeast Asia (SEAS).

To evaluate the performance of CMIP6 in simulating surface PM_2.5_ over Asia and its sub-regions, the root-mean-square error (RMSE) and the correlation coefficient with the observation are applied. The RMSEs and correlation coefficients are calculated for each CMIP6 model (or multimodel ensemble) over Asia and its six sub-regions.

To investigate the impact of future surface PM_2.5_ on population exposure, the populations in SSP1, SSP2, and SSP5 available in 2010–2100 are employed, with a horizontal resolution of 0.5° × 0.5° (interpolated to 1° × 1° for analysis). The change in population is considered.

The World Health Organization (WHO) safety standard for PM_2.5_ is less than 10 μg/m^3^. Thus, to investigate the impact of future changes in PM_2.5_ exposure on population in Asia, we calculated the time evolutions of the population exposed to the surface PM_2.5_ concentrations of greater than 10 μg/m^3^, population-weighted surface PM_2.5_ concentrations in different sub-regions, as well as the spatial distributions of population-weighted surface PM_2.5_ concentrations change in Asia in the early-, mid- and late-21st century.

The population-weighted surface PM_2.5_ concentrations is expressed as: $$POP\_weighted\_PM_{2.5}=PM_{2.5}\times POP/region\_ave\left(POP\right)$$
where *POP_weight_PM*_2.5_ is population-weighted surface PM_2.5_ concentrations, *PM*_2.5_ is surface PM_2.5_ concentrations, *POP* is population, and *region_ave (POP)* is the regional mean population.

### 2.4. Conversion from Density to Mass Mixing Ratio

Surface PM_2.5_ observations are the density dataset, while CMIP6 models output mass mixing ratio PM_2.5_ data. For comparison, PM_2.5_ density were conversed to mass mixing ratio using the formula as:$$mmr_{PM_{2.5}}=\frac{m_{PM_{2.5}}}{m_{air\;}-m_{PM_{2.5}}}\approx \frac{m_{PM_{2.5}}}{m_{air\;}}=\frac{\rho_{PM_{2.5}}\cdot V_{PM_{2.5}}}{\rho_{air\;}-V_{air}}$$
where $V_{air}=V_{PM_{2.5}}$, thus $mmr_{PM_{2.5}}\approx \;\frac{\rho_{PM_{2.5}}}{\rho_{air\;}}$.

Here, $mmr_{PM_{2.5}}$ is the mass mixing ratio of surface PM_2.5_ in a certain volume of air, and $m_{PM_{2.5}}$, $m_{air}$, $\rho_{PM_{2.5}}$, and $V_{PM_{2.5}}$ represent the PM_2.5_ mass, air mass, PM_2.5_ density, and PM_2.5_ volume in the corresponding volume of air, respectively. $\rho_{air\;}$ is air density and $V_{air}$ is air volume.

Air density is measured by the following formula:$$\rho_{air\;}=1.293\times \frac{P}{1000\;hPa}\times \frac{273.15}{\left(t+273.15\right)}$$
where *P* is air pressure and *t* is surface air temperature from CMIP6 models or observations. The PM_2.5_ density ($\rho_{PM_{2.5}}$) in each grid is the product of the mass mixing ratio of PM_2.5_ and the air density, with a unit of μg/m^3^.

## 3. Performance of CMIP6 GCMs

Figure 1 shows the spatial distributions of annual mean surface PM_2.5_ concentrations derived from the observations and the ensemble mean of CMIP6 models, as well as bias of CMIP6 models, during 1998–2014 over Asia. SEDAC shows high values of surface PM_2.5_ concentrations over Saudi Arabia, Iran, and Afghanistan in WAS, Kazakhstan, Uzbekistan, Kyrgyzstan, Turkmenistan, and Tajikistan in CAS, China in EAS, India and Pakistan in SAS. A maximum greater than 80 μg/m^3^ is mainly found over Saudi Arabia, India, Xinjiang Province of China, and eastern China, where the main source area of natural or anthropogenic emissions in Asia. The spatial pattern of MERRA-2 is consistent with that of SEDAC, but with a lower value over eastern China (Figure 1a,b). The ensemble mean of original CMIP6 GCMs output (ori-MME) well reproduces the spatial pattern of observed surface PM_2.5_ concentrations, with a high value (40 μg/m^3^) over Saudi Arabia, India, and eastern China. The correlation coefficient between ori-MME and SEDAC is 0.85, while the value is 0.84 when MERRA-2 is the reference. The spatial pattern of the estimated surface PM_2.5_ concentrations from the CMIP6 GCMs (com-MME) is in good agreement with MERRA-2 and SEDAC, with lower values than those from ori-MME. The correlation coefficient between com-MME and SEDAC is 0.93, while the value is 0.77 when MERRA-2 is the reference (Figure 1c,d). Compared with SEDAC, the biases of ori-MME and com-MME are mostly negative over Asia (Figure 1e,g). Compared with MERRA, by contrast, the biases of CMIP6 GCMs are positive over India and eastern China (Figure 1f,h). The RMSE of the ori-MME with the SEDAC is 15.42 μg/m^3^, with the largest value at the SAS (22.27 μg/m^3^) and the lowest value at the NAS (5.50 μg/m^3^). And the biases from the com-MME is similar, with an RMSE value of 17.33 μg/m^3^ at the whole Asia.

The individual models well reproduce the spatial pattern from SEDAC, but generally underestimate over NAS. Among the individual CMIP6 GCMs, IPSL-CM5A2-INCA underestimates over the whole of Asia, GISS-E2-1-G, GISS-E2-1-H, MIROC-ES2L, and MPI-ESM-1-2-HAM underestimate over northwestern Xinjiang in China, MIROC6 underestimates over WAS and NAS; while the spatial pattern from BCC-ESM1, GFDL-ESM4, MRI-ESM2-0, and NorESM2-LM exhibits good agreements with that from SEDAC. The correlation coefficient between individual models with the original simulated surface PM_2.5_ concentrations and the SEDAS range from 0.59 (MIROC-ES2L) to 0.90 (GISS-E2-1-G), while the values range from 0.53 (MIROC6) to 0.84 (GISS-E2-1-G) when MERRA-2 is the reference. Compared to the original simulated surface PM_2.5_ concentrations, the values of the estimated concentrations are higher than those from SEDAC. The values of ori-MME are close to MERRA-2. The largest bias is found in MRI-ESM2-0, followed by MIROC-ES2L and NorESM2-LM. The model uncertainty of surface PM_2.5_ concentrations is large (~20 μg/m^3^) over the source area of natural aerosols in India and China. The correlation coefficient between individual models with the estimated surface PM_2.5_ concentrations and the SEDAS range from 0.72 (NorESM2-LM) to 0.92 (IPSL-CM5A2-INCA), while the values range from 0.41 (MIROC6) to 0.81 (BCC-ESM1) when MERRA-2 is the reference. Overall, the performance of the MMEs are at the higher or medium levels compared to most individual models for both the whole Asia and its sub-regions (shown in Appendix A).

Figure 2 provides the model spread of original simulated surface PM_2.5_ concentrations. A good agreement among the models is found over northern Asia with low values (<10 μg/m^3^) of surface PM_2.5_ concentrations, whereas a large model spread is found over Saudi Arabia, Iran, and Xinjiang Province of China, with high values of surface PM_2.5_. It may be caused by the wide disparity in the representations of dust among CMIP6 models (Figure 2a). Compared to the original simulations, the estimated surface PM_2.5_ concentrations have a smaller model spread, with a more consistent spatial distribution (Figure 2b). In brief, the simulated surface PM_2.5_ concentrations are in good agreement, which is in line with previous studies based on global and regional climate models [45,46,47,48,49]. The bias of CMIP6 GCMs maybe caused by the following: the uncertainty of emission inventory (e.g., local source of dust), the bias of dry and wet deposition scheme, the deficiency in organic aerosol (e.g., secondary organic aerosol) formation mechanisms, and the low resolution of GCMs.

To evaluate the performance of CMIP6 models in simulating monthly surface PM_2.5_ concentrations in different regions of Asia, the annual cycles of the original and estimated surface PM_2.5_ in sub-regions of Asia based on MERRA-2 are compared in Figure 3. Surface PM_2.5_ concentrations peak in spring and summer over WAS, in spring and autumn over EAS, and in summer and autumn over SAS. Both ori-MME and com-MME capture the annual cycle of surface PM_2.5_ concentrations over Asia well, but with underestimates over CAS and WAS and overestimates over other sub-regions. The monthly differences simulated by CMIP6 GCMs are small over NAS and SEAS. The estimated annual cycles of surface PM_2.5_ concentrations over NAS, SAS, SEAS, and EAS are close to that in MERRA-2 but underestimated over WAS. Due to the sparsely covered distributed ground-based observation, we cannot investigate the source of the bias of CMIP6 models.

## 4. Projection of Surface PM_2.5_ in Asia

In this section, future changes in surface PM_2.5_ and its components under different SSPs are projected, using the estimated PM_2.5_ only.

Figure 4 presents the temporal evolutions of future surface PM_2.5_ concentrations simulated by CMIP6 GCMs under five SSPs over different sub-regions of Asia. In SSP1-2.6, SSP2-4.5, and SSP5-8.5, relative to the present day, annual mean surface PM_2.5_ concentrations will decrease in all sub-regions of Asia, with obvious differences among the scenarios. The magnitudes and timings of changes at the regional scale are quite different. The largest decrease is found in SAS, with a decrease value of 8 μg/m^3^ in SSP1-2.6 and 6 μg/m^3^ in SSP5-8.5 until 2100, which indicates the improvement of air quality in this sub-region. The similar result is found over EAS. The change in surface PM_2.5_ concentrations over NAS is small, due to the low concentrations of BC and dust, with relatively small model uncertainty.

In SSP3-7.0, which is a weak air pollution controls pathway, the annual mean surface PM_2.5_ concentrations in SAS are projected to increase. The value of surface PM_2.5_ concentrations at present in this sub-region is high and will further increase by 8 μg/m^3^ by 2050, which is consistent with the increases of BC, SO_4_, and OA (Appendix A). In SSP3-7.0, the annual mean surface PM_2.5_ concentrations over EAS will also increase by 2.5–2.7 μg/m^3^ until 2050. Future changes over SEAS are similar to those over EAS, with an increment of 2 μg/m^3^.

The typical lifetime of PM_2.5_ is less than 2 weeks in the troposphere, thus PM_2.5_ are widely referred to as short-lived climate forcers (SLCFs) [50,51]. Under SSP370-lowNTCF, strong emission controls for SLCFs are implemented on top of SSP3-7.0 scenario. In this pathway, surface PM_2.5_ concentrations will decline immediately around 2030, with marked declines in SAS, EAS, and SEAS.

Under SSP1-2.6, SSP2-4.5, and SSP5-8.5, surface PM_2.5_ concentrations over Asia in all seasons will decrease, with the largest reductions in SAS, WAS, and SEAS. Among four seasons, winter witnesses the biggest declines, having a decline of 12 μg/m^3^ over SAS under SSP1-2.6 in the late-21st century. Under SSP3-7.0, the increments in winter are the largest over all sub-regions, except for NAS, followed by those in autumn, spring, and summer; the rise in SAS (larger than 10 μg/m^3^ in winter until 2050) is the biggest among all sub-regions. Under SSP370-lowNTCF, downward trends in surface PM_2.5_ concentrations are found in all seasons over all sub-regions of Asia; the values of SAS in all seasons peak around 2030, followed by a rapid decline.

The sign of trends in surface PM_2.5_ and its component changes are consistent. The BC concentrations grow prominently over SAS, WAS, and EAS in SSP3-7.0, with weak air controls occurring, while fall gradually in other SSPs. Among all seasons, the largest increase in BC occurs in winter under SSP3-7.0 (Appendix A). For dust, the large interannual variability is found in WAS, CAS, SAS, and EAS, chiefly because it is a kind of natural emission; the upward trends are found over CAS and WAS under all SSPs, with the biggest increases in spring and summer; a downward trend occurs over EAS; values of NAS and SAS remain (Appendix A).

The annual mean SO_4_ concentrations show a downward trend over most sub-regions/under most SSPs. But under SSP3-7.0 and SSP5-8.5, those in SAS will increase before 2050, with the largest rise in autumn and winter, and then decrease gradually after 2050 (Appendix A). SS is mainly distributed over on the coast and is projected to increase slightly over SAS and SEAS, with no obvious trend in WAS, EAS, and NAS. Among all seasons, the biggest increases over WAS, SAS, and SEAS are found in summer, followed by those in autumn and winter (Appendix A). Major future changes in OA mainly occur in SAS, SEAS, and EAS, with changes randomly occurring in other sub-regions. The largest increases in autumn and winter are projected under SSP3-7.0, which is consistent with the changes in BC. Under other SSPs, a remarkable downward trend in OA is found (Appendix A).

Figure 5 displays the spatial distributions of the projected changes in the annual mean estimated surface PM_2.5_ concentrations during 2021–2040, 2041–2060, and 2081–2100 in five SSPs, based on the ensemble mean of CMIP6 GCMs. Under SSP1-2.6, surface PM_2.5_ concentrations are projected to decrease over eastern Asia in all time periods, with the biggest declines over SEAS and SAS in the late-21st century. In SSP2-4.5, surface PM_2.5_ concentrations over SAS will increase in the early-21st century and decrease in the mid- and late-21st century. The reduction in winter is larger than that in summer (Appendix A).

Under SSP3-7.0, the area close to natural aerosol emission source (SAS) has the biggest future changes in the annual mean surface PM_2.5_ concentrations, with the largest increase (~30 μg/m^3^) in winter. The increasing magnitudes in winter are larger than those in summer over most parts of SAS, which indicates the obviously seasonal variation characteristics simulated by CMIP6 GCMs. In summary, the decline in aerosol and its precursor leads to the decrease in surface PM_2.5_ concentrations over most parts of Asia, which is in line with the results of CMIP5 models [52]. It is good for air quality improvement and human health. Under SSP370-lowNTCF, the reduction of SLCF will help to ameliorate air quality.

Under SSP1-2.6, SSP2-4.5, and SSP5-8.5, surface BC concentrations will decrease substantially over EAS at all time intervals, especially in the late-21st century. It is the main cause of declines in winter and summer surface PM_2.5_ concentrations over EAS. But in SSP3-7.0, the increments of BC concentrations are the biggest over EAS and SAS, with a larger increase in winter over SAS than that over EAS (Appendix A).

The changes in dust vary in different sub-regions of Asia. Increases of dust are mainly distributed in CAS and WAS under SSP1-2.6 and in WAS and northern EAS under SSP2-4.5. Changes in SSP5-8.5 are consistent with those in SSP2-4.5. Decreasing trends are found over most parts of Asia in SSP3-7.0 and SSP370-lowNTCF, which are opposite to the trends in PM_2.5_. This indicates the relatively small contribution of dust concentrations to PM_2.5_ concentrations. Among all seasons, increases in spring and summer are more obvious (Appendix A).

The changes in the spatial patterns of annual mean OA are consistent with those of BC, with marked decreasing trends in the mid- and late-21st century under SSP1-2.6, SSP2-4.5, and SSP5-8.5. The increases over SAS and EAS are projected in the mid-21st century under SSP3-7.0. OA concentrations will increase over part of NAS under SSP2-4.5, SSP3-7.0, SSP370-lowNTCF, and SSP5-8.5 (Appendix A).

Annual mean SO_4_ shows the upward trends in the early-21st century are found in all SSPs but decreases considerably in the late-21st century in SSP1-2.6, SSP2-4.5, and SSP5-8.5. The largest increase is found over SAS in winter (Appendix A).

Under all SSPs, SS over the land area of Asia will decline. The SS will increase in the Pacific north of 15°N in SSP1-2.6 and will decrease in the ocean in SSP2-4.5 and SSP5-8.5 (Appendix A).

In summary, surface PM_2.5_ concentrations will decrease, due to the decline of BC, SO_4_, OA, and dust.

## 5. The Impact of Surface PM_2.5_ over Asia on Population Exposure

Figure 6 provides the time evolutions of the population exposed to the surface PM_2.5_ concentrations of greater than 10 μg/m^3^. The population exposure in different sub-regions varies in different SSPs. In NAS, the population exposure presents a downward trend in three SSPs. It can be explained by the small population and the relatively low PM_2.5_ concentrations of NAS. The population exposure in EAS will increase before 2050, then decrease, with the largest downward trend in SSP1-2.6 and an increase in the late-21st century under SSP5-8.5. In WAS and CAS, population exposures will continue to grow, and decrease slightly under SSP2-4.5. After 2050, downward trends are found in all sub-regions under SSP1-2.6 and SSP5-8.5, with the largest trend in SAS. In SEAS, population exposures will decline in SSP1-2.6 and SSP2-4.5; in SSP5-8.5, it shows a sudden rise after 2050 and a rapid decline around 2085. The highest values of population exposed to the surface PM_2.5_ concentrations of greater than 10 μg/m^3^ are found in SEAS and SAS, followed by EAS and WAS, with the lowest values in NAS.

Figure 7 shows the time evolutions of population-weighted surface PM_2.5_ concentrations. Under all three SSPs, the decreasing trends in EAS, NAS, and SEAS are projected. It indicated that annual mean exposure to surface PM_2.5_ will be improved over most parts in the future. But population-weighted surface PM_2.5_ concentrations present a gradually increasing trend in WAS, which may be due to the increases of dust over this region. Although there is a gradual decrease in SAS after 2040, the amount of PM_2.5_ in this sub-region is larger than that in other sub-regions, which is linked to the large population in India.

Figure 8 displays the spatial distributions of population-weighted surface PM_2.5_ concentrations in the early-, mid- and late-21st century under three SSPs. In all SSPs, the highest values (>400 μg/m^3^) of surface PM_2.5_ concentrations exist in SAS and the eastern part of EAS, followed by Saudi Arabia (~100 μg/m^3^), and the values in NAS, CAS, and the Tibetan Plateau are less than 50 μg/m^3^. In the early-21st century, the spatial patterns of surface PM_2.5_ concentrations are consistent under all SSPs. In the mid-21st century, an area with high values reduces in SSP1-2.6, but the patterns in SSP2-4.5 and SSP5-8.5 remain. In the late-21st century, an area with high values shows obvious reductions in SAS and eastern EAS in all SSPs, which indicates a decline in the population exposed to the pollution caused by surface PM_2.5_.

## 6. Conclusions

Based on CMIP6 GCMs, this paper evaluated the performance of simulating the historical surface PM_2.5_ and its components, and then projected the future changes in surface PM_2.5_ and its components as well as their impact on population exposure. Our key conclusions can be summarized as follows:(1)CMIP6 models reproduce the spatial pattern of surface PM_2.5_ concentrations quite well. The original outputs of CMIP6 models underestimate surface PM_2.5_ concentrations, with a large/small model uncertainty over the areas with low/high climatic mean values of PM_2.5_ concentrations (NAS/Saudi Arabia, Iran, and Xinjiang Province of China). The model spread of the estimated concentrations is smaller than that from the original simulation. Generally, CMIP6 GCMs well capture the observed annual cycles and seasonal variations of surface PM_2.5_ concentrations in Asia.(2)Relative to the present day (1995–2014), surface PM_2.5_ concentrations are projected to decrease in SSP1-2.6, SSP2-4.5, and SSP5-8.5. The magnitudes and timings of changes at the regional scale are quite different. The sub-region with the sharpest fall is South Asia (SAS). Under SSP3-7.0, the increase of surface PM_2.5_ concentrations in SAS is the largest, with 8 μg/m^3^ increment by 2050; under SSP370-lowNTCF, the largest decreases of surface PM_2.5_ concentrations exist in SA, EAS, and SEAS. The downward trends in all seasons are consistent with that in annual trends under SSP1-2.6, SSP2-4.5, and SSP5-8.5, and the largest decreases are found in winter, with the reduction of 12 μg/m^3^ in SAS under SSP1-2.6 at the end of 21st century. Contrasting with the changes in other SSPs, surface PM_2.5_ concentrations will increase in winter under SSP3-7.0, with the biggest increment in SAS. Under SSP370-lowNTCF, surface PM_2.5_ concentrations in all sub-regions of Asia will decline.(3)In the weak air pollution controls pathway (SSP3-7.0), the annual mean BC concentrations are projected to increase markedly in SAS, WAS, and EAS. The concentrations will decrease gradually in other SSPs. Annual and seasonal mean dust concentrations show the obvious interannual variation over WAS, CAS, SAS, and EAS. Downward trends of SO_4_ are found in most parts of Asia under most SSPs. Larger changes in OA are found mainly in SAS, SEAS, and EAS. SS mainly distributes in coastal areas, with a gradual increasing trend over SAS and SEAS.(4)Among all SSPs, the substantial decreases in surface PM_2.5_ concentrations over the whole of Asia in all time intervals are found under SSP1-2.6, with the largest declines in EAS and SAS in the late-21st century (2081–2100). Under SSP2-4.5, surface PM_2.5_ concentrations will increase in the early-21st century and decrease in the mid- and late-21st century; the extent in winter is larger than that in summer. Under SSP3-7.0, the area close to natural aerosol emission source (SAS) has the biggest future changes in the annual mean surface PM_2.5_ concentrations, with the largest increase (~30 μg/m^3^) in winter. The obviously seasonal variation characteristics are simulated by CMIP6 GCMs.(5)The population exposure in different sub-regions varies in different SSPs. The population exposure in EAS will increase before 2050, then decrease, with the largest downward trend in SSP1-2.6. In WAS and CAS, the population exposure will continue to grow, and decrease slightly under SSP2-4.5. After 2050, the downward trends are found in all sub-regions under SSP1-2.6 and SSP5-8.5, with the largest trend in SAS. In SEAS, the population exposure will decline in SSP1-2.6 and SSP2-4.5; in SSP5-8.5, it shows a sudden rise after 2050 and a rapid decline around 2085.(6)The population-weighted surface PM_2.5_ concentrations show downward trends over CAS, EAS, NAS, and SEAS and an increasing trend over WAS under three SSPs. Although they gradually decrease in SAS after 2040, the amount of PM_2.5_ in this sub-region is larger than that in other sub-regions, which is linked to the large population in India.

## 7. Discussion

The results from CMIP6 provide an opportunity to assess the simulation of historical and future changes in air pollutants within the latest generation of Earth system and climate models using up-to-date scenarios of future socioeconomic development. Different changes in air pollutants were simulated over the historical period and future. Future regional concentrations of PM_2.5_ will vary depending on the climate and mitigation assumptions. Model diversity across the CMIP6 models is largest near the dust source regions due to their sensitivity to meteorological variability. Disagreements in the projection of future changes to regional surface PM_2.5_ concentrations between individual CMIP6 models could be attributed to differences in the complexity of the aerosol schemes implemented within models, in particular the formation mechanisms of organic aerosols and emission of natural precursor over certain regions, along with the magnitude of the climate change signal (temperature and precipitation) simulated by models and also the impact related to natural aerosol emissions via Earth system couplings. Further research and understanding of these processes are necessary to improve the robustness of regional projections of PM_2.5_ on climate change timescales.

In conclusion, surface PM_2.5_ concentrations will decrease due to the decline of aerosol and its precursors, which is in line with the results of CMIP5 models. This is good for air quality improvement and human health. If no controls to air pollution emissions in Asia with rapid economic development are taken, surface PM_2.5_ concentrations will increase, which will lead to worse air quality. Under SSP370-lowNTCF, the reduction of SLCF will help to improve air quality.

## Figures and Tables

**Figure 1 ijerph-19-12092-f001:**
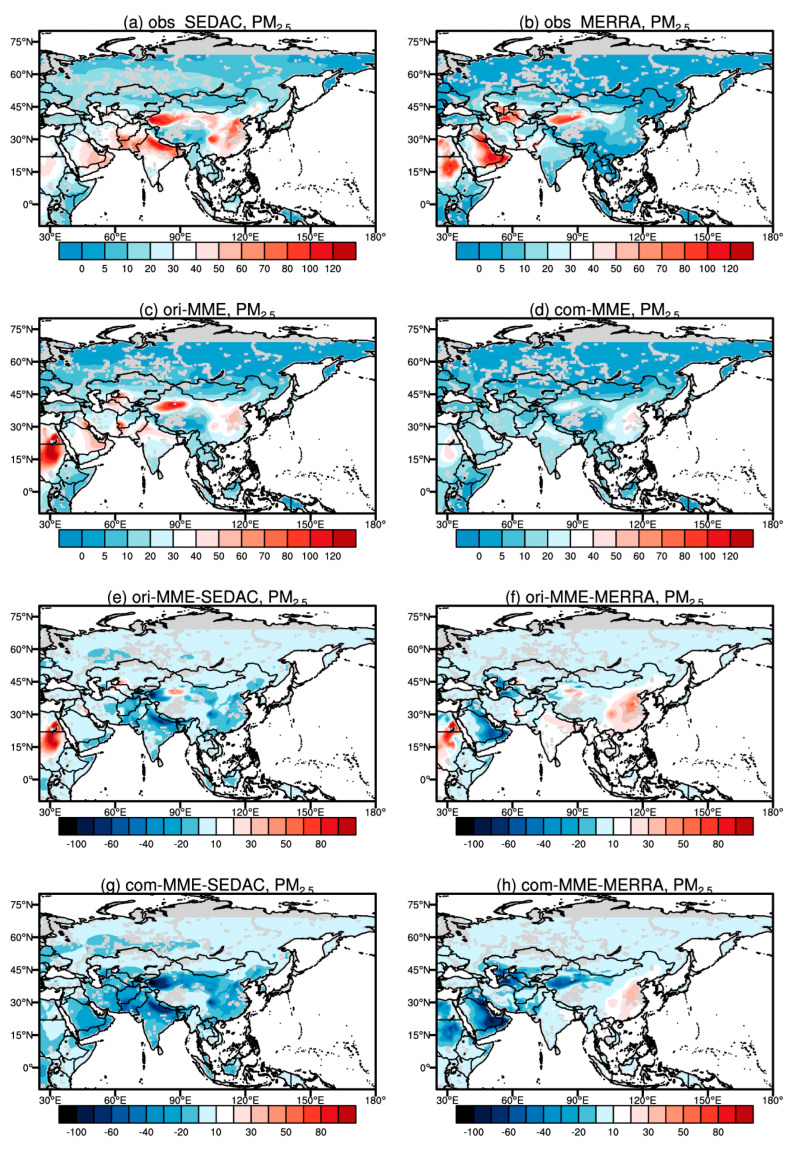
Spatial distribution of annual mean surface PM_2.5_ concentrations (units: μg/m^3^) during 1998–2014 from the ensemble mean of models and their biases.

**Figure 2 ijerph-19-12092-f002:**
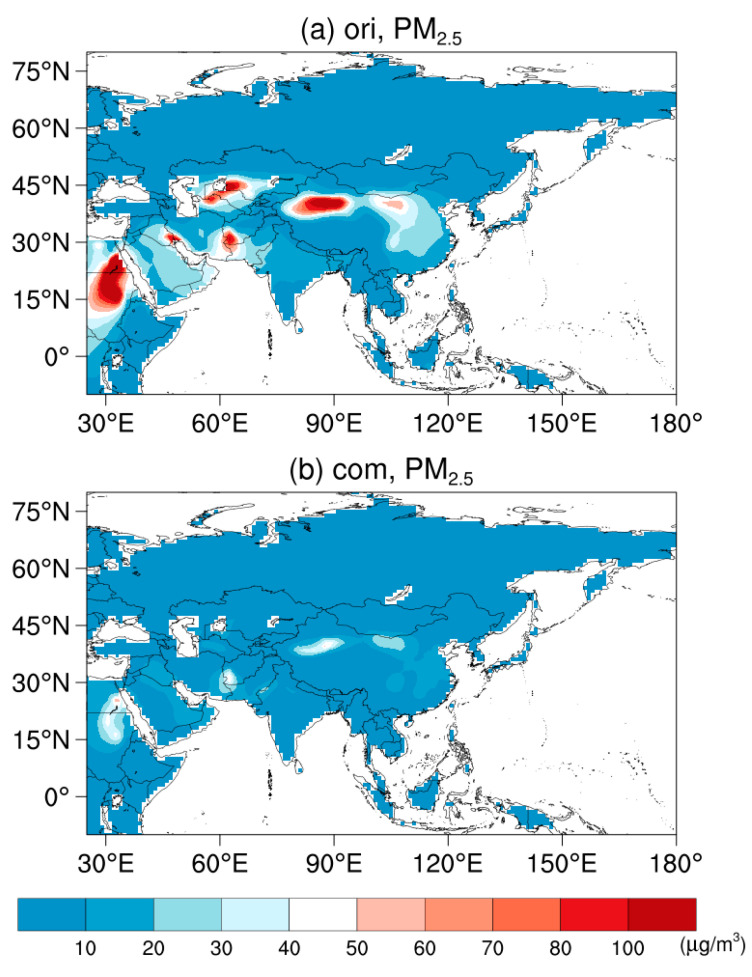
The standard deviation of simulated surface PM_2.5_ concentrations (units: μg/m^3^) among models during 1998–2014 from (**a**) the original simulations and (**b**) the estimations.

**Figure 3 ijerph-19-12092-f003:**
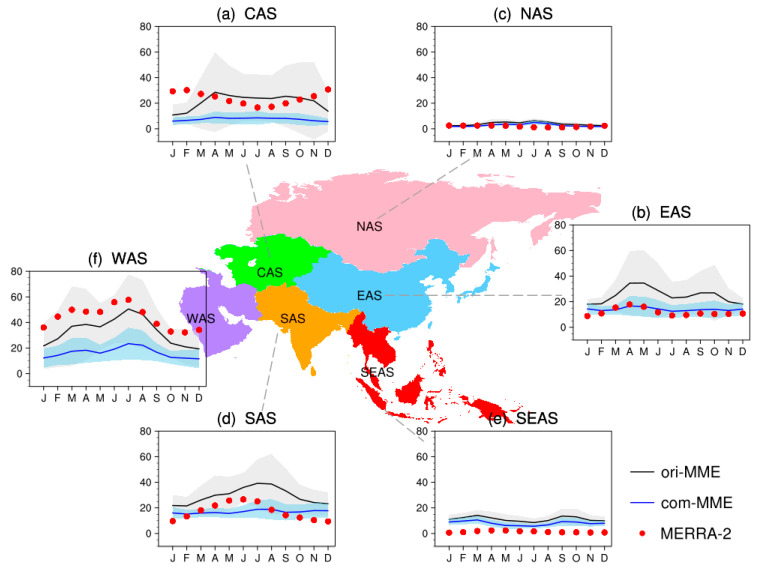
Monthly mean surface PM_2.5_ concentrations (units: μg/m^3^) over Asia during 1998–2014 from MERRA-2 (red dots), the original simulation (black curves), and the estimation (blue curves), where shadings indicate model uncertainty range (1 standard deviation among models).

**Figure 4 ijerph-19-12092-f004:**
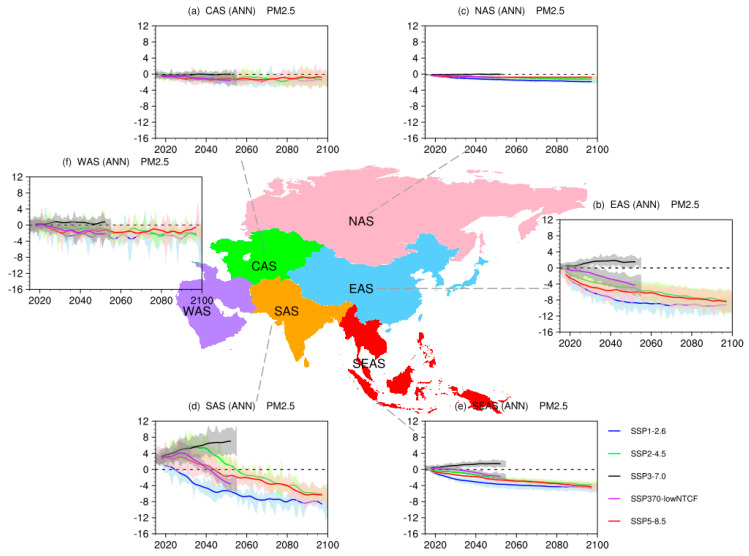
Time evolutions of annual mean surface PM_2.5_ concentrations (units: μg/m^3^) over Asia under SSP1-2.6 (red curves), SSP2-4.5 (green curves), SSP3-7.0 (black curves), SSP370-lowNTCF (purple curves), and SSP5-8.5 (red curves), relative to 1995–2014, where shadings indicate model uncertainty range (1 standard deviation of the simulations).

**Figure 5 ijerph-19-12092-f005:**
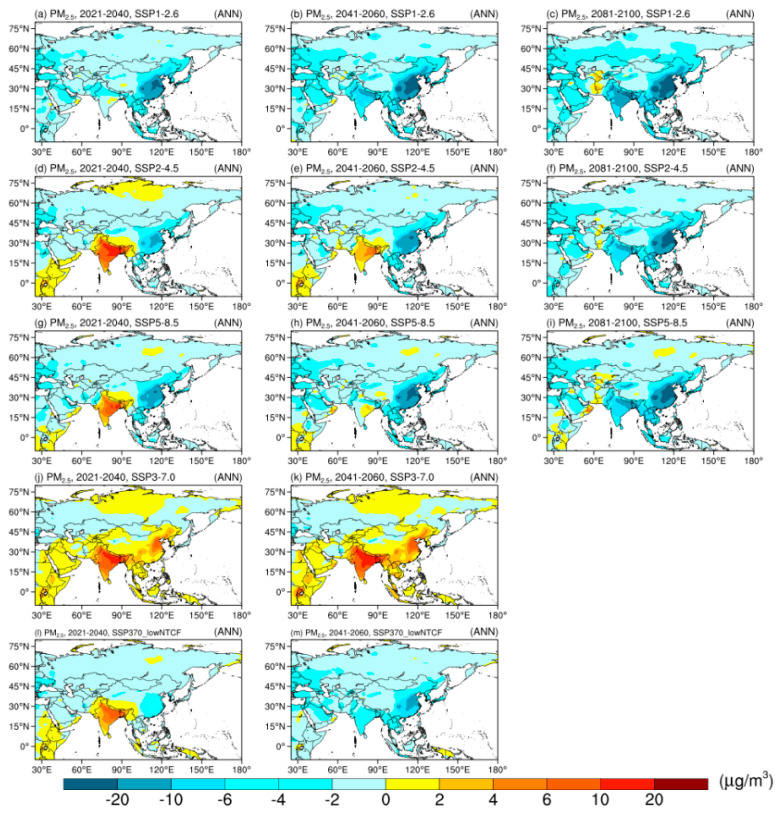
Spatial distribution of projected changes in surface PM_2.5_ concentrations annually (units: μg/m^3^) during 2021–2040, 2041–2060 and 2081–2100 under SSPs, relative to 1995–2014.

**Figure 6 ijerph-19-12092-f006:**
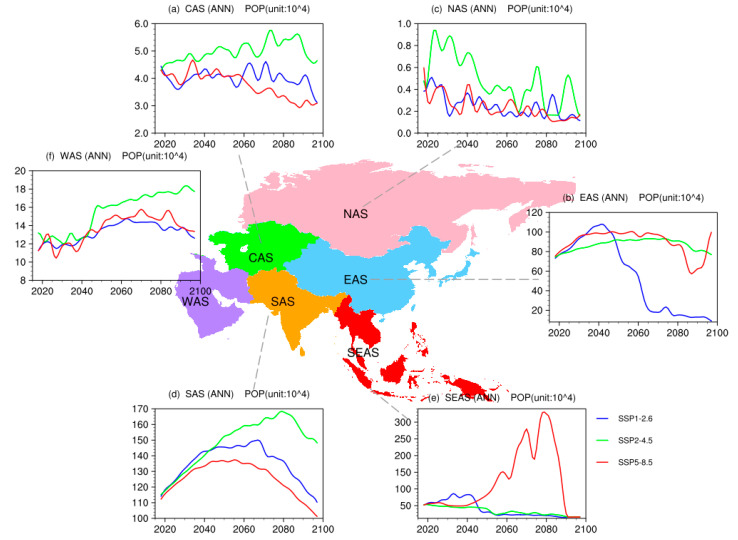
Time evolutions of the population (units: 10^4^) exposed to the surface PM_2.5_ concentrations of greater than 10 μg/m^3^ over Asia under SSP1-2.6 (blue), SSP2-4.5 (green), and SSP5-8.5 (red).

**Figure 7 ijerph-19-12092-f007:**
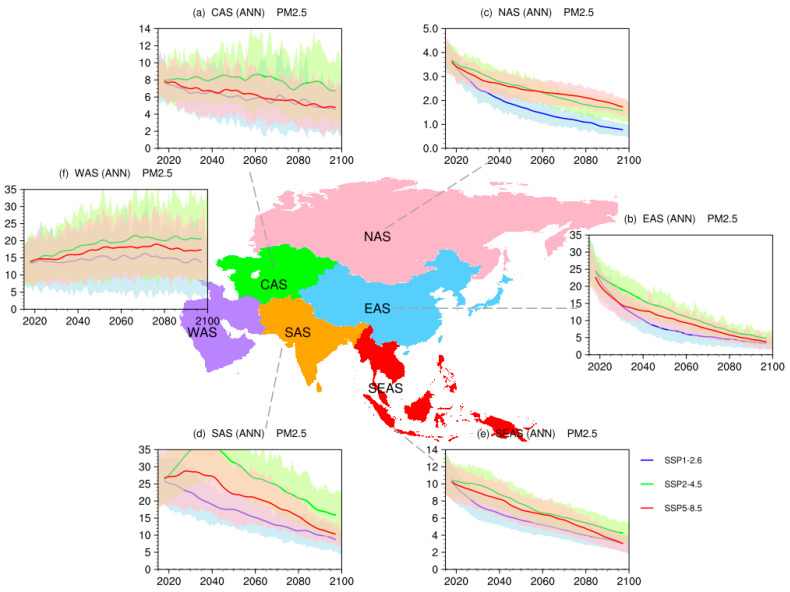
Time evolutions of population-weighted annual mean surface PM_2.5_ concentrations (units: μg/m^3^) over Asia under SSP1-2.6 (blue), SSP2-4.5 (green), and SSP5-8.5 (red), where shadings indicate model uncertainty range (1 standard deviation of the simulations).

**Figure 8 ijerph-19-12092-f008:**
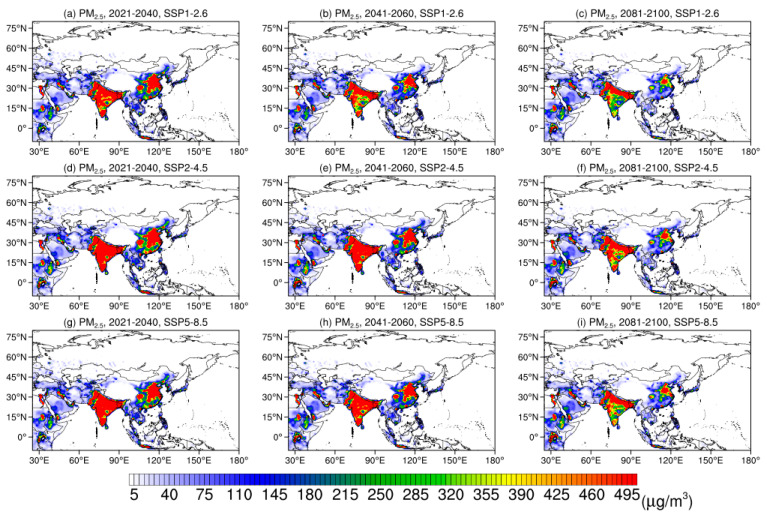
Spatial distribution of population-weighted annual mean surface PM_2.5_ concentrations (units: μg/m^3^) during 2021–2040, 2041–2060, and 2081–2100 under SSP1-2.6, SSP2-4.5, and SSP5-8.5.

**Table 1 ijerph-19-12092-t001:** List of CMIP6 models used in this study.

Model	Present Day	SSP1-2.6	SSP2-4.5	SSP3-7.0	SSP370-lowNTCF	SSP5-8.5	Horizontal Resolution (Longitude × Latitude)	Institute and Country; Data Citation
BCC-ESM1	*			*	*		2.813° × 2.813°	Beijing Climate Center (BCC), China; [32,33]
GFDL-ESM4	√ *	√ *	√ *	√ *	√ *	√ *	1.0° × 1.25°	Geophysical Fluid Dynamics Laboratory (GFDL), United States; [34,35]
GISS-E2-1-G	√ *	√		√ *	√ *	√	2.5° × 2°	Goddard Institute for Space Studies (GISS), United States; [36]
GISS-E2-1-H	√ *	√	√ *	√ *	√ *	√ *	2.5° × 2°	Goddard Institute for Space Studies (GISS), United States; [36]
IPSL-CM5A2-INCA	*			*	*		3.75° × 2.813°	Institute Pierre-Simon Laplace (IPSL), France; [37,38]
MPI-ESM-1-2-HAM	*			*	*		1.875° × 1.875°	Max Planck Institute (MPI) for Meteorology, Germany; [39,40]
MIROC-ES2L	√ *	√ *	√ *	*		*	2.813° × 2.813°	The University of Tokyo, Japan; [41]
MIROC6	√ *	√ *	√ *	√ *	*	√ *	1.406° × 1.406°	Model for Interdisciplinary Research onClimate (MIROC), Japan; [42]
MRI-ESM2-0	√ *	√ *	√ *	√ *	√ *	√ *	1.125° × 1.125°	Meteorological Research Institute (MRI),Japan; [43]
NorESM2-LM	√ *	√	√	√ *	√ *	√	1.9° × 2.5°	Norwegian Climate Centre, Norway; [44]
NorESM2-MM	√	√		√		√	0.9° × 1.25°	Norwegian Climate Centre, Norway; [44]

Note: √ represents a model output PM_2.5_ only, * represents a model output black carbon (BC), organic aerosol (OA), SO_4_, dust, and sea salt (SS).

## Data Availability

Not applicable.

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
