# Peer review of "Evaluation and Projection of Surface PM2.5 and Its Exposure on Population in Asia Based on the CMIP6 GCMs"

_ijerph, 2022, doi:10.3390/ijerph191912092_

Round 1
Reviewer 1 Report
This work explores the CMIP6’s simulation of PM2.5 with different scenarios, and a solid work has been done, which will for sure be beneficial to the community. However, the paper itself is not well-written. I am not saying the language, but the background knowledge, methodology and discussions are not well performed, in another word, the story line was not clear. Suggestions can be given like:
More information could be included in the introduction, such as the previous study about PM2.5 exposure in other regions, with other models like CMIP5, origin of the PM2.5 in Asia, etc;
The methodology section could have more information, which will be mentioned in the specific points below;
In the conclusion section we miss the discussion, like why there are different trends for different pollutants, how to explain the difference among the sub-regions, are there any suggestions from the authors etc.
More specific comments are:
Line 4: Is Jie Xu’s affiliation number 3?
Line 47: is this total number of one year or the accumulation since what time?
Line 56: State of the art?
Line 64: it is a bit weird to see SSP1-2.6 here since all SSPs will be introduced in the next section.
Line 130: is there any literatures supporting the factors used here?
Under Figure 2: the regions should be introduced and explain the abbreviations in the methodology section
Section 5: the population data used in this study as well as the method to calculate PM2.5 exposure should be introduced in the methodology part, while the WHO standard should be moved to either introduction or methodology section.
Figure 6: does this consider the change of population?
Figure 8: the definition/calculation of population weighted PM2.5 concentration should be explained in the methodology section.
Above figure 5: ‘in line with CMIP5’, here a reference is needed.
Author Response
This work explores the CMIP6’s simulation of PM2.5 with different scenarios, and a solid work has been done, which will for sure be beneficial to the community. However, the paper itself is not well-written. I am not saying the language, but the background knowledge, methodology and discussions are not well performed, in another word, the story line was not clear. Suggestions can be given like:
More information could be included in the introduction, such as the previous study about PM2.5 exposure in other regions, with other models like CMIP5, origin of the PM2.5 in Asia, etc;
Reply: Added in L54-L61.
The methodology section could have more information, which will be mentioned in the specific points below;
Reply: Thank you for comment. Replies can be found below.
In the conclusion section we miss the discussion, like why there are different trends for different pollutants, how to explain the difference among the sub-regions, are there any suggestions from the authors etc.
Reply: Changed.
More specific comments are:
Line 4: Is Jie Xu’s affiliation number 3?
Reply: Yes, changed in Line 4.
Line 47: is this total number of one year or the accumulation since what time?
Reply: The number is the total number of one year, “every year” has been added in Line 49.
Line 56: State of the art?
Reply: Yes, changed in Line 64.
Line 64: it is a bit weird to see SSP1-2.6 here since all SSPs will be introduced in the next section.
Reply: “SSP1-2.6” has been deleted.
Line 130: is there any literatures supporting the factors used here?
Reply: Turnock et al. (2020) and Su et al. (2022) support the factors used here, these literatures have been added in Line 130.
Under Figure 2: the regions should be introduced and explain the abbreviations in the methodology section
Reply: The regions have been introduced and explain the abbreviations in Lines 143-146.
Section 5: the population data used in this study as well as the method to calculate PM2.5 exposure should be introduced in the methodology part, while the WHO standard should be moved to either introduction or methodology section.
Reply: Moved to methodology section (Lines 156-163).
Figure 6: does this consider the change of population?
Reply: Yes. We added “The change of population is considered” in Lines 154-155.
Figure 8: the definition/calculation of population weighted PM2.5 concentration should be explained in the methodology section.
Reply: Thanks your suggestion, added in the methodology section, please see Lines 164-167.
Above figure 5: ‘in line with CMIP5’, here a reference is needed.
Reply: Added.
Reviewer 2 Report
The authors evaluated the performance of CMIP6 in simulating historical surface PM2.5 in Asia, and then projected surface PM2.5 to evaluate its changes and their impact on population exposure under various SSP scenarios. Although the topic is good, there are many unexplained objects that need to be further improved.
-
In the introduction: The authors go to great lengths to provide a lot of background, including the harmful effects of air pollution, the applicability of the latest CMIP6 models, and the importance of the study area. However, the entire introduction section makes no mention of the innovation of this research or its distinction from previous studies. More comprehensive comparisons between the present study and the other studies should be addressed. Please indicate the innovation of your study and the shortcomings of previous studies at the end of Introduction.
-
When the authors evaluated the model simulation results of CMIP6, they did not use any evaluation metrics to make a scientific judgement, but only used plots with naked eye judgement. This is not the most desirable approach. The authors need to use RMSE, MAE, or other evaluation metrics to measure the deviation of the simulation results from the observation. Please consider adding these to make results more reliable.
-
Appropriate citations are required when presenting the calculation formulas. Otherwise, the calculation results will not be reliable.
-
In the sentence:“The trends in surface PM2.5 and its components are consistent.”. How you assessed this trend is not explained here.
-
In the sentence:“A typical lifetime of PM2.5 is less than 2 weeks in the troposphere, thus PM2.5 is widely referred to as short-lived climate forcers (SLCFs).” There are no words that explain how this lifetime is calculated. Please explain it.
-
In the sentence:“Thus, to investigate the impact of future changes in PM2.5 exposure on population in Asia, we calculated the time evolutions of the population exposed to the surface PM2.5 concentrations of greater than 10 , population-weighted surface PM2.5 concentrations in different sub-regions, as well as the spatial distributions of population-weighted surface PM2.5 concentrations change in Asia in the early-, mid- and late21st century”. Please describe the calculation process or equation in concise language and add appropriate citations to the literature.
-
Some sentences have obvious grammar errors. The authors should improve the English further.
Author Response
The authors evaluated the performance of CMIP6 in simulating historical surface PM2.5 in Asia, and then projected surface PM2.5 to evaluate its changes and their impact on population exposure under various SSP scenarios. Although the topic is good, there are many unexplained objects that need to be further improved.
- In the introduction: The authors go to great lengths to provide a lot of background, including the harmful effects of air pollution, the applicability of the latest CMIP6 models, and the importance of the study area. However, the entire introduction section makes no mention of the innovation of this research or its distinction from previous studies. More comprehensive comparisons between the present study and the other studies should be addressed. Please indicate the innovation of your study and the shortcomings of previous studies at the end of Introduction.
Reply: More information on previous studies have been added in the introduction section. Please see L54-L61.
- When the authors evaluated the model simulation results of CMIP6, they did not use any evaluation metrics to make a scientific judgement, but only used plots with naked eye judgement. This is not the most desirable approach. The authors need to use RMSE, MAE, or other evaluation metrics to measure the deviation of the simulation results from the observation. Please consider adding these to make results more reliable.
Reply: More evaluations based on RMSE and correlation coefficient are given. See Section 3.
- Appropriate citations are required when presenting the calculation formulas. Otherwise, the calculation results will not be reliable.
Reply: Added “following Turnock, et al. [15] and Su, et al. [16]” in Line 138.
- In the sentence:“The trends in surface PM2.5 and its components are consistent.”. How you assessed this trend is not explained here.
Reply: What we assessed here is the sign of trends, but not the magnitude of trends. This sentence has been changed to “The sign of trends in surface PM2.5 and its components changes are consistent”. For example, in SEAS, the downward trends of PM2.5, BC and OA can be found under SSP1-2.6, SSP2-4.5, SSP370-lowNTFC and SSP5-8.5, with the upward trends of PM2.5, BC and OA under SSP3-7.0.
- In the sentence:“A typical lifetime of PM2.5 is less than 2 weeks in the troposphere, thus PM2.5 is widely referred to as short-lived climate forcers (SLCFs).” There are no words that explain how this lifetime is calculated. Please explain it.
Reply: The lifetime is the global atmospheric lifetime, which characterises the time required to turn over the global atmospheric burden. It is defined as the burden divided by the mean global sink for a gas in steady state. However, the PM2.5 compounds cannot be uniformly mixed throughout the troposphere, so the values of lifetimes should be viewed as approximations (Prather et al., 2001). The value of lifetimes of PM2.5 components come from several previous studies (Table R1). The Chapter 6 of IPCC AR6 WGI report has assessed these studies, and all the values of lifetimes we mentioned are shown in this report (Szopa et al., 2021).
Table R1. The source types, lifetime in the atmosphere of the PM2.5 compounds, which comes from the Table 6.1 of Chapter 6 in the IPCC AR6 WGI report
Compounds |
Lifetime |
Carbonaceous Aerosols (BC and OA) |
Minutes to Weeks |
Sea spray (SS) |
Day to week |
Sulphate aerosols (SO4) |
Minutes to weeks |
Dust |
Minutes to Weeks |
Prather, M.J. et al., 2001: Atmospheric Chemistry and Greenhouse Gases. In: Climate Change 2001: The Physical Science Basis. Contribution of Working Group I to the Third Assessment Report of the Intergovernmental Panel on Climate Change [Y. Ding, D.J. Griggs, M. Noguer, P.J. van der Linden, X. Dai, K. Maskell, and C.A. Johnson (eds.)]. Cambridge University Press, Cambridge, United Kingdom and New York, NY, USA, pp. 239–287, www.ipcc.ch/report/ar3/wg1.
- In the sentence:“Thus, to investigate the impact of future changes in PM2.5 exposure on population in Asia, we calculated the time evolutions of the population exposed to the surface PM2.5 concentrations of greater than 10 , population-weighted surface PM2.5 concentrations in different sub-regions, as well as the spatial distributions of population-weighted surface PM2.5 concentrations change in Asia in the early-, mid- and late21st century”. Please describe the calculation process or equation in concise language and add appropriate citations to the literature.
Reply: Added in L164-L.167
- Some sentences have obvious grammar errors. The authors should improve the English further.
Reply: Thank you for comment. We have polished the English in this paper thoroughly.

Reviewer 3 Report
Manuscript ijerph-1853482
Title: Evaluation and Projection of Surface PM2.5 and Its Exposure on Population in Asia Based on the CMIP6 GCMs
Authors: Ying Xu *, Jie Wu, Zhenyu Han
Submitted to section: Water Science and Technology
This study evaluates the historical (1995-2014) PM2.5 concentrations in Asia and based on the Global Climate Model CMIP6 the authors predicts future surface changes in PM2.5 and their exposure on the population (2010-2100). After evaluating the models with the historical data, the authors predict that PM2.5 concentrations will decrease in the future due to the decline of BC, SO4, OA, and dust. An extend of changes will depend on the region and season. The authors claim that the population exposure in different sub-regions will vary in the future depending on shared socio-economic pathways. The study is well conducted and well written and all the elements of a manuscript properly presented. This is an important subject as it deals with the most-populated regions in the world with the highest mortality to predict and understand better the future changes of PM2.5 and the impacts on the population exposure.
Minor comments:
Lines 29, 33: “it indicates that annual mean population exposure to surface PM2.5 will be smaller.”, “the surface PM2.5 concentrations over the most area of Asia will decrease” - how about the population number for the same time and region, will it also decrease or increase? – usually there is a positive correlation between population and pollution. It is hard to say if the “exposure to surface PM2.5 will be smaller” if do not compare PM levels with the population number.
Line 56: “includes the start-as-the-art climate”, did the authors mean state-of-the-art?
Line 83: “In accordance with the Paris Agreement for keeping global temperature below 2 °C,”, unclear 2o below what, please explain.
Line 107: “The period of 1998-2014 is selected for evaluation.”, please explain here why not 1998-2019?
Line 133: “The whole of Asia is the target region for analysis is.”, unclear.
Author Response
This study evaluates the historical (1995-2014) PM2.5 concentrations in Asia and based on the Global Climate Model CMIP6 the authors predicts future surface changes in PM2.5 and their exposure on the population (2010-2100). After evaluating the models with the historical data, the authors predict that PM2.5 concentrations will decrease in the future due to the decline of BC, SO4, OA, and dust. An extend of changes will depend on the region and season. The authors claim that the population exposure in different sub-regions will vary in the future depending on shared socio-economic pathways. The study is well conducted and well written and all the elements of a manuscript properly presented. This is an important subject as it deals with the most-populated regions in the world with the highest mortality to predict and understand better the future changes of PM2.5 and the impacts on the population exposure.
Minor comments:
Lines 29, 33: “it indicates that annual mean population exposure to surface PM2.5 will be smaller.”, “the surface PM2.5 concentrations over the most area of Asia will decrease” - how about the population number for the same time and region, will it also decrease or increase? – usually there is a positive correlation between population and pollution. It is hard to say if the “exposure to surface PM2.5 will be smaller” if do not compare PM levels with the population number.
Reply: The population in the future will decrease. In the meantime, the concentration of pollutant will decrease with the implementation of emission reduction measures. Thus, population-weighted PM2.5 concentration will decrease. “it indicates that annual mean population exposure to surface PM2.5 will be smaller” has been changed to “it indicates that the surface PM2.5 concentrations and population over the most area of Asia will decrease” in L29-L30.
Line 56: “includes the start-as-the-art climate”, did the authors mean state-of-the-art?
Reply: Yes, we have changed it in L64.
Line 83: “In accordance with the Paris Agreement for keeping global temperature below 2 °C,”, unclear 2o below what, please explain.
Reply: Thank you for the comment. This sentence has been changed to “In accordance with the Paris Agreement for keeping global temperature below 2 °C of global warming above pre-industrial levels” in L92-L93.
Line 107: “The period of 1998-2014 is selected for evaluation.”, please explain here why not 1998-2019?
Reply: Because the coverage of historical simulations from CMIP6 is 1850-2014. “Considering the temporal coverage of SEDAC, MERRA-2 and historical simulations of CMIP6” has been added in L115-L116 to explain.
Line 133: “The whole of Asia is the target region for analysis is.”, unclear.
Reply: This sentence has been changed to “The whole of Asia is the target region for analysis” in L143.
Round 2
Reviewer 2 Report
The authors have revised the manuscript accordingly, which is good. However, some obvious grammatical errors should be further corrected.
1. In the sentence: “To comparison, observed PM2.5 density data were conversed to mass mixing ratio data using the formula as: ”. Please carefully check the grammar of this sentence.
2. It seems that there is an article usage error problem here. For instance, “in a good agreement with”, “with the high values”, “with the lower values”, “with the largest value at the SAS (22.27 ) and lowest value at”. Please change these examples and other similar expressions into the correct ones.
3. In the sentence: “Although the gradually decrease in SAS after 2040, the amount of PM2.5 in this sub-region is larger than that in other sub-regions, which is links to the large population in India.” Consider change “the gradually decrease” into “the gradual decease”, and modify “is links to” as “is linked to”.
4. In the sentence: “which lead to the worse of the air quality. Under SSP3-7.0-lowNTCF, the reduction of SLCF will helps to improve air quality.” Consider change “the worse of the air quality” into “worse air quality”, and modify “will helps” as “will help”.
Author Response
- In the sentence: “To comparison, observed PM2.5 density data were conversed to mass mixing ratio data using the formula as: ”. Please carefully check the grammar of this sentence.
Reply: Changed to “To comparison, PM2.5 density were conversed to PM2.5 mass mixing ratio using the formula as”.
- It seems that there is an article usage error problem here. For instance, “in a good agreement with”, “with the high values”, “with the lower values”, “with the largest value at the SAS (22.27 ) and lowest value at”. Please change these examples and other similar expressions into the correct ones.
Reply: Thank you for the comment. Changed to “in good agreement with”, “with a high value”, “with lower values”, “with the largest value at the SAS (22.27 ) and the lowest value at”. We have examined and changed similar expressions thoroughly.
- In the sentence: “Although the gradually decrease in SAS after 2040, the amount of PM2.5 in this sub-region is larger than that in other sub-regions, which is links to the large population in India.” Consider change “the gradually decrease” into “the gradual decease”, and modify “is links to” as “is linked to”.
Reply: Thank you for the comment. Changed.
- In the sentence: “which lead to the worse of the air quality. Under SSP3-7.0-lowNTCF, the reduction of SLCF will helps to improve air quality.” Consider change “the worse of the air quality” into “worse air quality”, and modify “will helps” as “will help”.
Reply: Thank you for the comment. Changed.